# Scaling up the primary health integrated care project for chronic conditions in Kenya: study protocol for an implementation research project

Ellen Nolte [1], Jemima H Kamano,[2] Violet Naanyu,[3] Anthony Etyang,[4] Antonio Gasparrini,[5] Kara Hanson,[6] Hillary Koros,[7] Richard Mugo,[7] Adrianna Murphy,[1] Robinson Oyando,[8] Triantafyllos Pliakas,[5] Vincent Were,[8] Ruth Willis,[6] Edwine Barasa,[9] Pablo Perel[10]

EN and JHK contributed equally. EB and PP contributed equally.

For numbered affiliations see end of article.

**Correspondence to**
Dr Ellen Nolte;
Ellen.Nolte@lshtm.ac.uk

## ABSTRACT

**Introduction** Amid the rising number of people with non-communicable diseases (NCDs), Kenya has invested in strengthening primary care and in efforts to expand existing service delivery platforms to integrate NCD care. One such approach is the AMPATH (Academic Model Providing Access to Healthcare) model in western Kenya, which provides the platform for the Primary Health Integrated Care Project for Chronic Conditions (PIC4C), launched in 2018 to further strengthen primary care services for the prevention and control of hypertension, diabetes, breast and cervical cancer. This study seeks to understand how well PIC4C delivers on its intended aims and to inform and support scale up of the PIC4C model for integrated care for people with NCDs in Kenya.

**Methods and analysis** The study is guided by a conceptual framework on implementing, sustaining and spreading innovation in health service delivery. We use a multimethod design combining qualitative and quantitative approaches, involving: (1) in-depth interviews with health workers and decision-makers to explore experiences of delivering PIC4C; (2) a cross-sectional survey of patients with diabetes or hypertension and in-depth interviews to understand how well PIC4C meets patients' needs; (3) a cohort study with an interrupted time series analysis to evaluate the degree to which PIC4C leads to health benefits such as improved management of hypertension or diabetes; and (4) a cohort study of households to examine the extent to which the national hospital insurance chronic care package provides financial risk protection to people with hypertension or diabetes within PIC4C.

**Ethics and dissemination** The study has received approvals from Moi University Institutional Research and Ethics Committee (FAN:0003586) and the London School of Hygiene & Tropical Medicine (17940). Workshops with key stakeholders at local, county, national and international levels will ensure early and wide dissemination of our findings to inform scale up of this model of care. We will also publish findings in peer-reviewed journals.

## Strengths and limitations of this study

► The study uses a multimethod approach combining qualitative (in-depth interviews), and quantitative approaches (patient surveys, cohort study with interrupted time series analysis and cohort study of households) to assess the implementation and potential for scale up of integrated primary care for chronic conditions in western Kenya.

► It involves longitudinal data collection to capture changes in experiences among decision-makers, healthcare workers and patients with chronic conditions over time and the extent to which insurance cover protects people with non-communicable diseases (NCDs).

► The study will be the first to use a large data set based on electronic health records to evaluate of the health benefits of an innovative model for the management of NCDs in sub-Saharan Africa.

► The PIC4C (Primary Health Integrated Care Project for Chronic Conditions) model of care is implemented in two counties in western Kenya only and it will be difficult to derive generalisable findings for all of Kenya.

► The use of electronic health records for research purposes can be fraught in that variables that are key for answering the research question may be missing and/or, for variables for which data is collected, there might be missing information.

## INTRODUCTION

Like many other countries in sub-Saharan Africa, Kenya faces the double burden of infectious and non-communicable diseases (NCDs).[1 2] In 2015, NCDs accounted for almost one-third of all deaths and half of hospital admissions; about a quarter of the population had hypertension and 5% had diabetes or impaired fasting glycaemia.[3] Knowledge about NCDs and their risk factors varies widely, as do access to and use of related services, with the poor remaining especially vulnerable.[4 5] People with NCDs are more likely to incur catastrophic health

expenditure.[6] The Kenyan government has successively expanded the benefits package of the National Hospital Insurance Fund (NHIF), although less than one-fifth of the population is covered.[7] The 'Supa Cover' scheme includes a comprehensive set of services targeting NCDs but levels of uptake are low.[8] While Kenya has invested in strengthening primary healthcare (PHC),[9] the system has remained hospital-centric, with long waiting times and reduced quality of care.[10] Lack of systematic screening and early detection of NCDs, along with lack of capacity at lower levels of care, means that a substantial number of people are diagnosed at advanced disease stages requiring specialist input and further increasing pressure on hospitals and their staff.

However, there is clear commitment to re-balance service delivery from curative to preventative and integrated care to manage NCDs more effectively,[9] with examples of novel care models that have extended HIV services into the community to serve as a platform for NCD care integration.[9] One such model is the Academic Model Providing Access to Healthcare (AMPATH) in western Kenya.[11] Developed as a partnership between Moi University in Kenya and North American academic medical centres in 2001, it became one of sub-Saharan Africa's largest HIV treatment and control programmes, and from 2010, was progressively expanded to provide NCD care, serving a population of more than 8 million people at over 800 Ministry of Health facilities in 2020.[12] Building on these experiences, the Kenyan Ministry of Health launched, in 2018 and in partnership with AMPATH/Moi, Access Accelerated and the World Bank, the Primary Health Integrated Care Project for Chronic Conditions (PIC4C) (box 1).[13] PIC4C aims to strengthen primary care services for the prevention and control of hypertension, diabetes,

breast and cervical cancer. It is being piloted in two counties in western Kenya, and accompanied by a small-scale evaluation involving a before-and-after community and facility survey, a limited process evaluation and a costing study.[14] The study described here extends the ongoing evaluation to understand how well PIC4C delivers on its intended aims and to inform and support scale up of the PIC4C model for integrated NCD management in Kenya.

### Research objectives
The study seeks to:
1. Assess the key components of the implementation process of the PIC4C model;
2. Understand the experiences of patients to assess whether and how well the PIC4C model meets the needs of those affected by the selected NCDs;
3. Assess the health benefits and potential unintended consequences of the implementation of the PIC4C pilot; and
4. Evaluate the effectiveness of the NHIF chronic care benefit package to provide financial risk protection, its responsiveness to the needs of individuals and impacts on equity, efficiency and quality of care.

### Conceptual framework
The project is guided by a conceptual framework that draws on our work on implementing, sustaining and spreading innovation in health service delivery.[15 16] It posits that ensuring that service innovation, such as the PIC4C model, is sustained, spread and scaled up, requires supportive and committed leadership and management at different tiers; early and widespread stakeholder involvement, including staff and service users; dedicated and ongoing resources, including funding, infrastructure, staff and time; effective communication across and between organisations; adaptation of the innovation to the local context; ongoing monitoring and timely feedback about progress; and evaluation and demonstration of (cost-)effectiveness. Our project considers all these factors to produce the evidence needed to support the wider spread and potential scale up of the PIC4C model.

---

> **Box 1  The Primary Health Integrated Care Project for Chronic Conditions (PIC4C) model of care**
>
> The PIC4C model aims to identify people with hypertension, diabetes, cervical and/or breast cancer in the community and ensure their timely referral to, treatment and management at the appropriate service level (health centre, dispensary, subcounty or county hospital). Implementation in Busia and Trans Nzoia counties commenced in February 2018.
>
> PIC4C targets different tiers of the delivery system and includes infrastructural measures. Specific components include (i) training of community health workers to undertake screening, prevention, referral and health education for hypertension, diabetes and cancer; (ii) providing equipment for screening, laboratory testing and treatment of cervical cancer and breast cancer; (iii) strengthening the supply chain; (iv) health records systems strengthening, building on the AMPATH (Academic Model Providing Access to Healthcare) Medical Record System;[28] (v) training and mentorship of nurses, clinical officers and general practitioners working in primary healthcare facilities to deliver evidenced-based management and referral; (vi) linkage to care using structured referral pathways between levels of care and with the voluntary Supa Cover operated by the National Hospital Insurance Fund for sustainable health financing.

---

## METHODS/DESIGN
This implementation research project uses a multimethod design combining qualitative (sequential in-depth interviews with health workers, decision-makers and patients), and quantitative approaches (patient surveys, a cohort study with an interrupted time series analysis and a cohort study of households (table 1). Figure 1 illustrates the overall study design in relation to the guiding framework. The study period is September 2019 to August 2022.

### Settings and site selection
The Kenyan public health system is organised into four tiers and six levels: (i) community health services (level 1) include all community-based activities guided by the Ministry of Health (MOH); (ii) primary care includes

**Table 1** Components of the study

| Study objective and subobjectives | Rationale / hypothesis | Data collection or source | Sampling | Analysis |
|---|---|---|---|---|
| **Understanding the implementation process**<br>▲ To assess the quality of leadership and management; levels of stakeholder involvement; adequacy of support mechanisms and resources; ability to adapt the intervention locally; and quality of communication and of monitoring and feedback | Core to the successful adoption and implementation as well as sustaining of organisational change associated with the introduction of the PIC4C model are the various stakeholders affected by the change, their understanding and acceptance and resultant commitment and buy-in to the proposed changes, in particular by front-line staff.[16] Organisational innovations such as the PIC4C are unlikely to succeed long-term if they fail to take into account the diverse patterns of interests, values and power relationships between those involved in the development, implementation and delivery of the new service model.[29] | In-depth interviews with<br>1. Health workers<br>2. Decision-makers. at three time points each. | ***Health workers***: approximately 40–50 health workers from different levels of service delivery selected from among the 73 health facilities in Busia (n=20–25) and Trans Nzoia (n=20–25) counties participating in PIC4C.<br>***Decision-makers***: approximately 25 decision-makers, including senior health executives and managers of range of health facilities in Busia and Trans Nzoia, purposively sampled to represent different governance levels. | Thematic analysis. |
| **Understanding patient experience**<br>▲ To assess whether and how well the PIC4C model meets the needs of those affected by diabetes and hypertension | There are widely documented challenges of retaining people positively screened for a given NCD in care and adhering to treatment in Kenya.[30 31] The reasons for this include financial difficulties[5] with more recent evidence pointing to the role of 'treatment burden' for adherence to treatment and quality of life.[32] Assessing this burden provides a useful lens to explore the degree to which the PIC4C model supports people with NCDs in managing their condition and is able to identify and target subgroups of patients at risk of poor outcomes because of lack of capacity and resources to engage in self-care and treatment. | 1. Patient survey of experiences with treatment and self-management (PETS).<br>2. In-depth interviews with patients at three time points. | ***PETS***: Random sample that is representative of people with hypertension and/or diabetes in terms of age, gender and broad socioeconomic status; we seek to arrive at a total sample of around n=300. This sample size was used by the developers of the PETS survey instrument to test its validity.[17]<br>***Patient interviews***: Sampled from patients who have responded to the PETS, aiming to interview 15–20 patients in each county (total sample: 30–40 patients), with attention to recruiting those who may face challenges in accessing services as identified from PETS. | ***PETS***: Descriptive statistics.<br>***Patient interviews***: Thematic analysis. |
| **Evaluation of health benefits**<br>▲ To identify individual-level factors associated with levels of hypertension, diabetes and HIV viral suppression<br>▲ To identify facility-level factors associated with levels of hypertension, diabetes and HIV viral suppression<br>▲ To evaluate temporal trends and the impact of scaling up the PIC4C model on health benefits and potential unintended consequences | The introduction of the PIC4C model will lead to a significant improvement in the management of hypertension and diabetes without adversely affecting HIV management (as measured using HIV viral suppression levels). Primary outcomes, treated as continuous variables, will differ according to diagnosis: systolic blood pressure (hypertension), fasting glucose and HbA1c (diabetes). In patients with more than one diagnosis all relevant outcomes will be considered. Secondary outcomes will include overall cardiovascular risk (as estimated by the WHO Risk Score,[33] treated as continuous variable, hypertension control (<140/90), diabetes control (fasting blood sugar (FBS)<8.0 mmol/L, random blood sugar (RBS)<10 mmol/L) and HIV viral suppression (<1000 copies/mL) treated as binary variables. We further seek to assess a range of process outcomes, such as guideline adherence; treatment initiation, adherence and retention; referral; lifestyle recommendations; number of diagnostic tests performed; and, at facility level; drug availability. | Cohort study of patients with hypertension, diabetes and/or HIV/AIDS. | Estimated sample size of 8000 patients with hypertension, 1000 with diabetes and 1000 with diabetes and hypertension required to detect a reasonable and relevant impact of the PIC4C model for all outcomes (90% power) based on the assumptions: systolic blood pressure change of 5 mm Hg (SD 15) ICC 0.05, HbA1c change 0.37% (SD 1.1%) ICC 0.04.[34] | ***Subobjectives 1 & 2***: Descriptive statistics.<br>***Subobjective 3***: Patient visit-level analysis: Directed Acyclic Graphs informed linear mixed-effects models for continuous and generalised mixed-effect models for binary outcomes.<br>Facility-level analysis: Interrupted time-series analysis. |

**Table 1** Continued

| Study objective and subobjectives | Rationale / hypothesis | Data collection or source | Sampling | Analysis |
|---|---|---|---|---|
| **Assessment of effectiveness of NHIF chronic care benefit package**<br>▲ To measure the effectiveness of the NHIF national scheme benefit package to provide financial risk protection to individuals with chronic diseases<br>▲ To examine the extent to which the NHIF national scheme benefit package is responsive to the needs of individuals with hypertension, diabetes, cervical and breast cancer<br>▲ To examine how the provider incentives generated by provider payment arrangements of the NHIF national scheme benefit package influence equity, efficiency and quality of care | Enrolment in a health insurance scheme does not always translate into expanded access to healthcare and financial risk protection.[35] Also, purchasing arrangements, specifically the relationship between purchasers (such as the NHIF) and healthcare providers as well as provider payment mechanisms can influence provider behaviour in a way that shapes the equity, efficiency, and quality of service provision.[36] Understanding these aspects of the NHIF national scheme benefit package will provide evidence about its scalability as a financial risk protection mechanism for households with members with a chronic disease. It will further inform design refinements to help improve scalability of the programme. | 1. Cohort study of households with at least one member with hypertension, diabetes or both.<br>2. In-depth interviews (IDIs) with decision-makers, facility managers and healthcare workers.<br>3. Focus group discussions (FGDs) with patients with diabetes/hypertension and household heads | 1. Target sample of n=960 individuals, with n=480 enrolled in NHIF and n=480 not enrolled in NHIF; each subsample to include n=160 individuals with hypertension, n=160 with diabetes and n=160 with both conditions. Estimates based on detection of a 15% point difference in the proportion of catastrophic health expenditure (40% in the control group), a design effect of 1.2 and a two-sided alpha level of 0.05 (80% power and estimated 60% attrition); sample stratified by county and chronic condition.<br>2. Target sample of 54 IDIs across national, and county and health facility levels in Busia and Trans Nzoia.<br>3. Four FGDs (each with approximately 10–12 participants), two each in Busia and Trans Nzoia counties, covering rural and urban areas. | 1. Descriptive statistics; Generalised Estimating Equations with out-of-pocket as dependent and NHIF enrolment as independent variables; estimation of concentration curves and indices.<br>2. Thematic analysis. |

HbA1c, glycated haemoglobin; ICC, intraclass correlation coefficient; NCD, non-communicable disease; NHIF, National Hospital Insurance Fund; PETS, Patient Experience with Treatment and Self-Management; PIC4C, Primary Health Integrated Care Project for Chronic Conditions.

services provided by public dispensaries (level 2 facilities) and health centres (level 3); (iii) county referral services include first referral subcounty hospitals (level 4) and second referral county hospitals (level 5) that are managed by the county; the fourth tier includes national (tertiary) referral services (level 6).

The PIC4C model is implemented across 73 level 2–5 facilities in Busia and Trans Nzoia counties (table 2 and online supplemental appendix table A1).

## Data types and collection

The following data types will be collected: in-depth interviews with patients, health workers and decision-makers; focus group discussions with patients; a patient survey of experiences with treatment and self-management; patient data extracted from the AMPATH Medical Record System (AMRS), complemented by data extracted from health facility registers; and a household survey.

We will conduct two series of in-depth interviews with health workers and decision-makers. The first relates to *Objective 1* of the study and seeks to explore questions around the quality of leadership and management; perceptions of and attitudes to the PIC4C model; levels of stakeholder involvement; adequacy of support mechanisms and resources; ability to adapt the intervention locally; quality of communication and of monitoring and feedback; and perceived challenges in the day-to-day delivery of the new service model. It involves health workers (physicians, clinical officers, nurses, laboratory staff, community health workers and Community Health Volunteers), working at different levels of service delivery and with decision-makers at the different tiers of the health system in Busia and Trans Nzoia counties. Sampling will be carried out purposively. Interviews will be semi-structured, following a topic guide that is informed by existing work on implementation and specific to the target group (health worker; facility manager; decision-maker)

The second series relates to *Objective 4* and will explore questions around the appropriateness of the NHIF benefit package to meet the needs of patients with chronic conditions, as well as perceived adequacy of provider payment to deliver the expected services in a way that achieves the goals of equity, efficiency and quality. It will focus on specifically on decision-makers with a role in financing, including representatives of the NHIF at national and county levels, and their counterparts at the MoH and county health management, as well as managers of public and private health facilities.

### In-depth interviews with patients

To address *Objective 2*, we will carry out interviews with patients with diabetes and/or hypertension from different socio-demographic backgrounds and geographical location (distance to health facilities), among other factors as identified from the patient survey (see below). Interviews will use a semi-structured interview guide informed by the literature on treatment and self-management burden, while also exploring people's experiences of the health

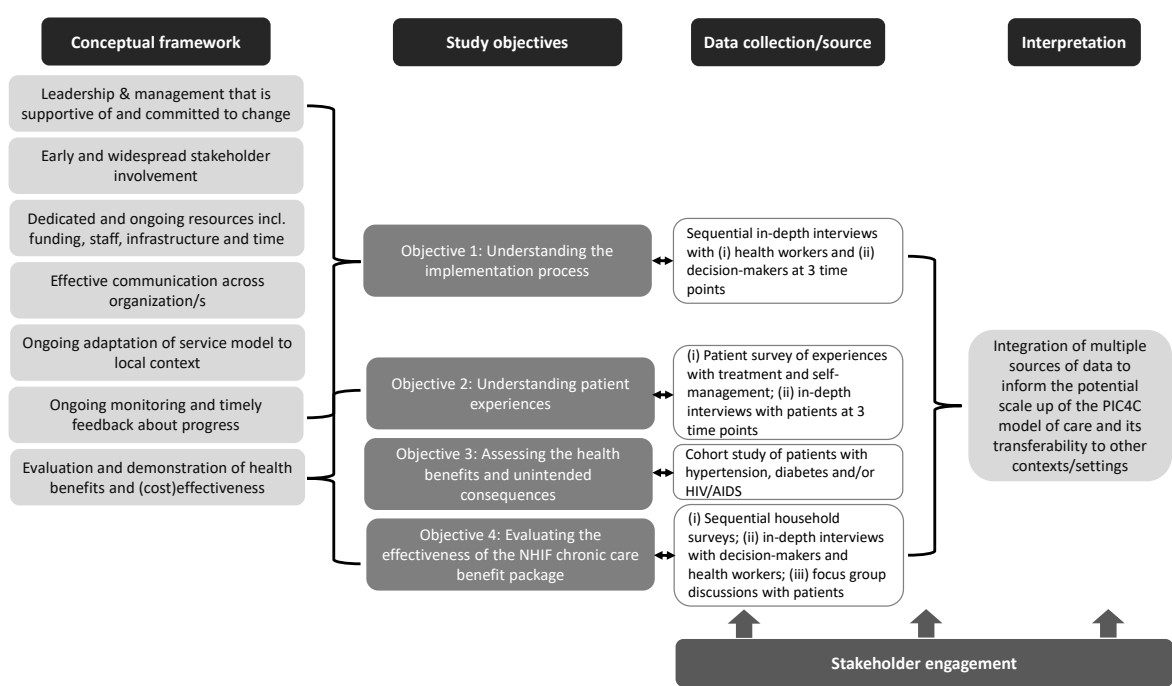

**Figure 1** Study design in relation to the guiding conceptual framework. incl., including; NHIF, National Hospital Insurance Fund; PIC4C, Primary Health Integrated Care Project for Chronic Conditions.

service, including how the PIC4C is making a difference to their experience. Interviews will seek to capture the experience of patients along the care pathway, with a focus on transition points between care levels; their level of involvement in treatment decisions; perceived quality of communication with the different types of health worker; and issues faced in terms of access and adherence to treatment.

All interviews will be conducted in English or Swahili, with topic guides translated (forward/backtranslation) and piloted with a small sample of the respective interview group. *Objective 1 and 2* interviews will be conducted at three points in time over the duration of the project; *Objective 4* interviews will be conducted at one point in time only. All interviews will be carried out face-to-face by researchers trained in qualitative methods in a location most convenient for the interview participant. Patient interviews will be carried out at a health facility closest to invited participants' home. Participants will be reimbursed to cover the cost of travel and will also be offered refreshments. We expect interviews to last between

45–60 min for health workers and decision-makers and up to 90 min for patients.

*Objective 4* further foresees focus groups discussions (FGDs) with patients. These will be conducted with households that are enrolled in NHIF and that have at least one member with hypertension and/or diabetes. FGDs will follow a topic guide and explore participants' experiences around accessing care, and their perceptions about the extent to which the NHIF national scheme meets their needs as patients with chronic disease. FGDs will be facilitated by researchers trained in qualitative methods.

## Patient survey of experiences with treatment and self-management

As part of *Objective 2*, we will conduct a survey of patients with hypertension and/or diabetes using the 'Patient Experience with Treatment and Self-Management' (PETS)[17] tool to understand their perceived treatment and self-management burden. We will test whether the questions in the current version of PETS are appropriate in the Kenyan context and add questions about the respondents' socio-demographic

| Table 2 | Number and type of facilities in Busia and Trans Nzoia counties participating in PIC4C | |
|---|---|---|
| | **Busia (*seven subcounties*)** | **Trans Nzoia (*five subcounties*)** |
| County hospital (*level 5*) | 1 | 1 |
| Subcounty hospital (*level 4*) | 7 | 5 |
| Health centre (*level 3*) | 10 | 13 |
| Dispensary (*level 2*) | 22 | 13 |
| Other | – | 1 (private non-profit medical centre) |
| **Total** | **40** | **33** |

background and health status. We will translate and back-translate the instrument into Swahili and test with a small number of patients. We will use computer-assisted personal interviewing with trained research assistants and the REDCap platform, a secure web application for building and managing online surveys and databases, which is already being used for data collection in the ongoing PIC4C evaluation. The survey will include an additional question asking participating respondents whether they would be happy to be contacted subsequently for an in-depth interview to further explore their experiences with treatment and self-management of their condition/s.

### Cohort study of patients with hypertension, diabetes and/or HIV/AIDS

The study population includes all adult (>18 years) patients with a diagnosis of hypertension, diabetes and/or HIV/AIDS. We will use the AMRS, in which each patient has been assigned a unique universal identifier and which is used in the management of their care across all levels. Primary outcomes are systolic blood pressure and fasting glucose or glycated haemoglobin; we will also collect data on HIV viral suppression (<1000 copies/mL), process outcomes and potential confounders at facility and individual level. Based on current AMRS data we estimate that approximately 8000 patients with hypertension, 1000 with diabetes and 1000 with diabetes and hypertension will fulfil the inclusion criteria and be included in the interrupted time-series analysis.

### Cohort study of households with at least one member with hypertension or diabetes

The study population includes households with at least one adult (≥18 years) member with hypertension and/or diabetes. Those enrolled with NHIF will be the exposed group and those without the control. Eligible households will be identified from the PIC4C AMPATH/Moi University screening database, which we will complement with (i) the wider AMRS database and/or (ii) health facility registers to identify households that meet our inclusion criteria. We will use a questionnaire administered by research assistants using computer-assisted personal interviewing based on the CommCare platform. We will conduct four rounds (follow-up at months 3, 6, 9 and 12). Respondents will include the head of the household and the household member with hypertension and/or diabetes, capturing healthcare seeking events, general household and healthcare expenditure, alongside household characteristics. The responding household member with hypertension/diabetes will also be invited to keep a diary recording healthcare seeking events and expenditure. The diary will be checked by data collectors to validate survey responses.

### Data analysis

#### In-depth interviews and focus group discussions

Interviews and focus group discussions will be recorded, with participants' permission, transcribed and translated into English. Interview transcripts will be delinked from

any personal information and allocated a unique identifier to ensure confidentiality. Interview analysis will use a thematic approach,[18 19] and include familiarisation with the data through reading and re-reading of transcripts and organising data by means of coding and re-coding of data through a series of reflexive steps. Codes will be initially generated from the interview guide and literature, with additional codes added as data are analysed. We will use NVivo software to assist with data management. Interview transcripts will be analysed for each participant group and round and comparisons drawn between the first, second and third round data to explore changes as a result of PIC4C implementation.

#### Patient survey of experiences with treatment and self-management

Survey data will be imported into Stata and analysed using simple descriptive statistics (continuous data will be presented as means and SD, and discrete data presented as frequencies and percentages) as the main aim is to understand the prevalence and patterns of the treatment and self-management burden experienced by people with hypertension and/or diabetes. This will inform recruitment for in-depth interviews with a smaller group of patients to gain further insight into the underlying issues to help explain the challenges experienced and so inform the further development and scale up of the PIC4C model.

#### Cohort study of patients with hypertension, diabetes and/or HIV/AIDS

Analyses will use individual (patient visit) level data and health services (facility) monthly aggregated level data. Descriptive analysis will identify individual-level and facility-level factors associated with levels of hypertension, diabetes and HIV viral suppression. We will generate a panel time series data set and describe the number of visits, of patients and reported outcomes within each facility at each month; we will also explore possible changes in the population composition over time and describe monthly proportions of patients achieving diabetes and hypertension control and be virally suppressed. We further seek to examine patient visit patterns, including the distribution of visits per patient and facility and to identify patients with at least one visit before and after the implementation of PIC4C to examine, in sensitivity analyses, the impact of PIC4C on patient outcomes. Descriptive analyses will take account of potential confounders and effect modifiers at patient and facility level informed by causal inference framework using directed acyclic graphs.[20]

Facility level analyses will use interrupted time series analysis using monthly facility level aggregated data to examine the impact of the PIC4C model on all outcomes. The main impact model will assume a stable level change after the PIC4C implementation which has been achieved progressively within the implementation period (June 2019 to November 2019). The time periods before and after the model implementation will constitute the

two segments of our regression models considering 24 monthly time points before and 24 monthly time points after. Analyses will consider an implementation period of 6 months in which the intervention is being rolled out. We will assess autocorrelation by examining the plot of residuals and the partial autocorrelation function and, perform Durbin-Watson test statistics to evaluate serial autocorrelation and adjust using autoregressive integrated moving average where necessary. Seasonality will be examined and accounted for in all models.

## Cohort study of households with at least one member with hypertension or diabetes

Analyses will include, first, summarising findings by computing annualised means and medians of health expenditure overall and key sample characteristics such as chronic condition. The primary outcome, level of out-of-pocket (OOP) health expenditure, will be computed as the proportion of total per capita household consumption expenditure. We will employ Generalised Estimating Equations to assess differences in level of OOP health expenditure between households enrolled in the NHIF and those that are not, using NIHF enrolment status as main independent variable. We will account for clustering at health facility level, applying the mixed effect of multilevel covariates, and conduct post-hoc matching of households on characteristics that may drive healthcare expenditure, such as the number of people in the households, gender, chronic condition, county of residence, socioeconomic status and number of older people living in the household. We will use the coarsened exact matching method, which seeks to control for the potential confounding of 'pre-treatment' covariates on the outcome of interest, by matching 'treatment' cases with 'non-treatment' cases that are approximately similar with regard to those covariates. In our case, 'treatment' cases are households enrolled in NHIF and 'non-treatment' controls are households that are not. The magnitude of the insurance effect will be measured by the adjusted values of coefficients of NHIF enrolment and significant results will be established at $p < 0.05$. The coefficients of the covariates will be converted into risk ratio or incidence rate ratios, as appropriate, with 95% CIs reported. We will also develop concentration curves and compute concentration indices of the level of OOP expenditure by categorising the households in the sample into five socio-economic quintiles (five richest and one poorest), using annual total household consumption expenditure as measure for household wealth. All statistical analyses will use Stata V.14.

## Patient and public involvement statement

The roll out of the PIC4C model in the two counties was accomplished by involving village chiefs and elders in the process. Patients and members of the public were not involved in the development of the proposal or the design of the research presented here.

## Ethical approval

Ethical approval has been granted for all elements of this study. Informed consent will be obtained from all participants in this study through their signatures on an informed consent form, in which participants confirm that they have read and understood the project information sheet. Additionally, at the beginning of each interview or the patient/household survey, the interviewer/survey administrator will seek verbal consent that participants are still willing to be interviewed/complete the survey and, in case of the interview, that they agree that it will be audio recorded and transcribed. All material will be made available in English or Swahili. In case of participants unable to read the information sheet and consent form, these will be read out to them in their native language and verbal consent will be requested and witnessed; where a suitable witness cannot be identified, the participant will not be interviewed.

All electronic data will be maintained in a secure password-protected environment and any hard copies of interview transcripts and consent forms will be kept in a locked cabinet. Once data have been processed and databases created, they will be encrypted, password-protected and saved in the Moi University secured network.

## Knowledge translation

We have strong established relationships with some of the key knowledge users (MoH, the World Bank and decision-makers at county levels) who are members of the study advisory committee and will be key to achieve the expected impact of the project. We will hold two formal workshops with key stakeholders directly and indirectly involved in PIC4C model development and implementation as well as national and county level political and health system leaders, academic and AMPATH leaders, senior level service providers, civil society and stakeholders from the other World Bank/Access Accelerated participating countries. Workshops aim to communicate emergent insights from the work; actively engage stakeholder and use their knowledge and experience to reflect on emergent findings; and formulate recommendations for a regional and/or national scale up strategy. They will also provide opportunity for adapting proposed work to ensure that the research meets the information needs of key stakeholders involved in the organisation and governance of the Kenyan health system at county and national levels. A final policy dialogue will be aimed at high-level national and international policymakers to disseminate the findings of the research to a broader policy audience with a view to inform implementation and scale up of integrated primary care approaches beyond Kenya.

## DISCUSSION

The burden of disease has changed globally, with a rise in NCDs posing an increasing challenge for low-income and middle-income countries where there remains emphasis on an acute, episodic model of care which is not well

suited to meet the requirements of people with NCDs and services are not well integrated across the different care levels. Strengthening PHC is widely accepted to be core to address the changing disease burden and a foundation for a sustainable health system.[21 22] Many countries in sub-Saharan Africa are exploring ways to strengthen existing PHC structures by integrating care for NCDs into routine HIV services.[23] However, while there are promising examples of innovative care models, these are often implemented as time-limited pilots, ending when project funding ends, or not extending beyond localised projects,[24] despite increasing evidence on the key factors facilitating scale up of innovations in low-resource settings.[25 26]

Our study will contribute to this evidence base by conducting a rigorous evaluation of an ambitious innovative integrated NCD care model. We will triangulate different methodological approaches and involve key stakeholders from the beginning and throughout all study phases. It will make a unique contribution to the emerging evidence and theory development on complexity around treatment and self-management burden among people with NCDs, which is a comparatively new field, in particular in low-resource settings.[27] By adopting a longitudinal approach, we will be able to capture changes in experiences among decision-makers, healthcare workers and patients with NCDs over time. In addition, the collection of household health expenditure data over the course of an entire year will enable overcoming seasonal biases that are typical for one-off expenditure surveys conducted elsewhere. Our study will still be the first to use a large data set based on electronic health records to evaluate the health benefits of an innovative model for the management of NCDs in sub-Saharan Africa.

Our approach to understanding PIC4C implementation emphasises depth over breadth by exploring the experiences of decision-makers, health workers and patients and how these change over time to help inform a possible roll-out of the care model to other parts of Kenya by means of in-depth interviews. These use, of necessity, smaller sample sizes which may not be representative of the population although we will seek to capture that range of views of those involved in or affected by PIC4C. At the same time, our household survey will likely provide us with a representative view of the degree to which the NHIF chronic care benefit package provides financial risk protection to individuals with chronic diseases. However, the survey will have to rely, to considerable extent, on self-reported expenditure, which is always susceptible to recall bias. We will seek to minimise potential bias by also including diaries alongside the survey instrument. Finally, while this study is in the fortunate position to draw on the well-developed AMRS, the use of electronic health records for research purposes can be fraught in that variables that are key for answering the research question may be missing and/or, for variables for which data is collected, there might be missing information. Notwithstanding these challenges, to best of our knowledge, our study will be the first to use a large data set based on electronic

health records for evaluating an innovative model for the management of NCDs in sub-Saharan Africa.

Overall, using a pragmatic, comprehensive and innovative methodological approach, our study will advance understanding of the integrated management of chronic conditions in low-resource settings. It will generate new knowledge about the key mechanisms and factors that shape the successful implementation of novel ways of working to deliver more integrated services to better manage and support people with chronic conditions, which will be relevant to settings beyond Kenya. It will provide important new insights into the requirements for scaling up a novel approach to managing NCDs in PHC from the perspective of those organising, delivering and receiving enhanced services while also expanding our understanding of the unintended consequences of integrating NCD management into the primary care platform of existing care programmes. This knowledge can then be leveraged to inform and improve the design and implementation of similar programmes elsewhere in Kenya and other low-resource settings.

**Author affiliations**
¹Department of Health Services Research and Policy, London School of Hygiene & Tropical Medicine, London, UK
²School of Medicine, Moi University College of Health Sciences, Eldoret, Kenya
³School of Arts and Social Sciences, Moi University, Eldoret, Kenya
⁴Department of Epidemiology and Demography, KEMRI-Wellcome Trust Research Programme, Kilifi, Kenya
⁵Department of Public Health, Environments and Society, London School of Hygiene & Tropical Medicine, London, UK
⁶Department of Global Health and Development, London School of Hygiene & Tropical Medicine, London, UK
⁷Academic Model Providing Access to HealthCare (AMPATH), Eldoret, Kenya
⁸Health Economics Research Unit, KEMRI-Wellcome Trust Research Programme, Kilifi, Kenya
⁹Health Economics Research Unit, KEMRI-Wellcome Trust Research Programme, Nairobi, Kenya
¹⁰Department of Non-communicable Disease Epidemiology, London School of Hygiene & Tropical Medicine, London, UK

**Acknowledgements** The authors wish to acknowledge the members of the Advisory Committee to the project for their valuable input into protocol development and the dissemination strategy.

**Contributors** PP, JHK, EN, VN, AM, AG, KH, AE and EB conceptualised the study and obtained funding. HK, TP, RO, RW and VW contributed to the methodological development and study design. RM will support data management and participant recruitment. HK, RO and VW will lead on all planned data collection. RW will co-lead the qualitative work and all authors will contribute to data analysis. EN drafted the manuscript. All authors read and approved the final manuscript.

**Funding** This work was supported by the UK Medical Research Council grant number MR/T023538/1.

**Competing interests** None declared.

**Patient consent for publication** Not applicable.

**Provenance and peer review** Not commissioned; externally peer reviewed.

terminology, drug names and drug dosages), and is not responsible for any error and/or omissions arising from translation and adaptation or otherwise.

**ORCID iD**
Ellen Nolte http://orcid.org/0000-0002-2289-117X

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
