## [Reviewer comments · BMJ Open]

ARTICLE DETAILS

TITLE (PROVISIONAL)	Scaling up the Primary Health Integrated Care Project for Chronic Conditions in Kenya: Study Protocol for an implementation research project
AUTHORS	Nolte, Ellen; Kamano, Jemima; Naanyu, Violet; Etyang, Anthony; Gasparrini, Antonio; Hanson, Kara; Koros, Hillary; Mugo, Richard; Murphy, Adrianna; Oyando, Robinson; Pliakas, Triantafyllos; Were, Vincent; Willis, Ruth; Barasa, Edwine; Perel, Pablo

VERSION 1 – REVIEW

REVIEWER	Mahomed, Ozayr University of KwaZulu-Natal, Public Health MEDicine
REVIEW RETURNED	14-Aug-2021

GENERAL COMMENTS	Thank you for the opportunity to review this well written protocol that clearly details the mixed methods study in evaluating the model of care. My comments are as follows: 1. The protocol lends to itself to implementation science research and the study design should be framed in this context2. Objective 1- could be classified as a formative evaluation. I have reservations of how only key informant interviews with decision makers address the outcome of assessing implementation. In my understanding there should be an input process and output assessment that includes onsite evaluation at sampled sites3. In order to understand your conceptual framework, a pictorial depiction will be appreciated4. Assessment of sustainability of the intervention needs to be considered
---

REVIEWER	MUTABAZI, Jean Claude École de Santé Publique, Université de Montréal, Médecine sociale et préventive-Option: Santé Mondiale
REVIEW RETURNED	03-Nov-2021

GENERAL COMMENTS	This protocol entitled “Scaling up the Primary Health Integrated Care Project for Chronic Conditions in Kenya: Study Protocol for an implementation research project” has a potential has potential to set an example in both understanding/developing of the integrated interventions to deal with the rising NCDs and their implementation in the contexts of Africa or of other LMICs. This project will also be a good contribution to the limited literature on successfully integrated interventions in Africa, especially in the interface/collusion between infectious and non-communicable diseases. Upon project completion, the paper will be very
--

	interesting for readers and useful for researchers. English is clear. The key words seem to be appropriate. The title is suitable. The abstract is well written. Objectives are well-aligned. Methods are very well described. Discussion is scientifically sound and acceptable. However, few changes are suggested to make it more understandable and implementable protocol, especially to the general public. The following few suggestions/comments would be helpful:  1. Add a reference to line 3, page 6, behind NCDs regarding the double burden of diseases 2. It's better (for papers compared to reports or policy documents) to enter references inside before the points or commas at the end of every referenced statement. 3. Page 8, line 2 or somewhere else, A graphically or diagrammatically presented conceptual framework (or the adapted one) would portray well the project and all those factors you described. If you think you can quickly draw based on the description you put together, that it would be helpful otherwise leave it. 4. Page 8, line 5, though multiple method design is well described, I would however consider this to be a mixed methods study. The navigating between qual. And quant. data collection and analysis would just lead the conclusion of mixed methods that are nowadays most used in most of integrated interventions and research projects. 5. Page 15, lines 14-25 I like this knowledge translation section! 6. Discussion mentioned the expected outcomes in terms of integrated management of NCDs within PHC in Kenya and beyond (pages 17-18), I would suggest to add a paragraph or a few lines arguing how this would work in other LMICs with similar contexts or in the same situation. Looking forward to reading your paper after your research project is completed. Congratulations to the authors!!!
--	---

VERSION 1 – AUTHOR RESPONSE

Reviewer 1	
 1. The protocol lends to itself to implementation science research and the study design should be framed in this context 	We agree and indeed, the title of the manuscript already reflects this notion. We have also now amended the introductory sentence to the 'Methods/design' section (p. 8, line 6ff) as follows: "This implementation research project uses a multimethod design combining qualitative (sequential in-depth interviews with health workers, decision-makers, and patients), and quantitative approaches (patient surveys, a cohort study with an interrupted time series analysis and a cohort study of households"

2. Objective 1- could be classified as a formative evaluation. I have reservations of how only key informant interviews with decision makers address the outcome of assessing implementation. In my understanding there should be an input process and output assessment that includes onsite evaluation at sampled sites	Thank you for this comment. There are different definitions of formative research, but we do believe that our methodological description is appropriate. Objective 1 is seeking to assess the key components of the implementation process of the PIC4C model as we state in our paper and involves interviews with decision-makers and health care workers. As we argue in Table 1, our starting point are the various stakeholders who are affected by the changes related to the introduction of PIC4C. We know from the evidence (which we cite) that organisational innovations such as the PIC4C are unlikely to succeed long-term if they fail to take into account the diverse patterns of interests, values and power relationships between those involved in the development, implementation and delivery of the new service model. This is what work carried out under Objective 1 is seeking to understand and so inform options for sustaining and potential scaling of the care model. We have not changed the text.
3. In order to understand your conceptual framework, a pictorial depiction will be appreciated	We thank you for raising this point. The conceptual framework guiding our work derives from a comprehensive review of the key lessons that can be learned from the implementation science literature rather than providing a 'new' framework as such (Nolte 2018). Its key components, thus, reflect the main insights identified by studies of the implementation, sustaining and spreading of innovation in health service delivery and organisation and as such sought to guide data collection and analytical strategy in the current study. As requested we have now produced a visual (Figure 1) depicting the relationship between the conceptual framework, study objectives and data collectio.
4. Assessment of sustainability of the intervention needs to be considered	We absolutely agree with the Reviewer and sustainability forms one of the core components of our conceptual framework (p. 7, line 20ff).
Reviewer 2	
1. Add a reference to line 3, page 6, behind NCDs regarding the double burden of diseases	Thank you for this suggestion. We had added the following references illustrating this issue (addition highlighted in the main text p. 6, line 3):  • Ministry of Health. Kenya national strategy for the prevention and control of non-communicable diseases 2015 - 2020. Nairobi: Ministry of Health, 2015.

	 Gouda HN, Charlson F, Sorsdahl K, et al. Burden of non-communicable diseases in sub-Saharan Africa, 1990-2017: results from the Global Burden of Disease Study 2017. Lancet Glob Health 2019;7(10):e1375-e1387.
2. It's better (for papers compared to reports or policy documents) to enter references inside before the points or commas at the end of every referenced statement.	Thank you very much for this comment. We have followed BMJ Open guidance which states that "Reference numbers in the text should be inserted immediately after punctuation (with no word spacing)." We are seeking advice from the Journal Editors on this point.
3. Page 8, line 2 or somewhere else, A graphically or diagrammatically presented conceptual framework (or the adapted one) would portray well the project and all those factors you described. If you think you can quickly draw based on the description you put together, that it would be helpful otherwise leave it.	Thank you for this suggestion. We have now added a graphical representation of our study conceptual framework (see also response (3) to Reviewer 1).
4. Page 8, line 5, though multiple method design is well described, I would however consider this to be a mixed methods study. The navigating between qual. And quant. data collection and analysis would just lead the conclusion of mixed methods that are nowadays most used in most of integrated interventions and research projects.	Thank you very much for raising this point. We agree that the study design includes elements of a mixed-methods approach (especially Objective 2 and, so some degree, Objective 4) and each method will contribute to informing the potential scale up of the PIC4C model of care as well as its transferability to other contexts/settings. However, we believe that the study design overall does not quite meet what we would consider as mixed methods as described by for example Greene et al. (1989) or Creswell et al. (2011). We would therefore consider that the term 'multimethod design' our study best.
5. Page 15, lines 14-25 I like this knowledge translation section!	Thank you so much for your kind comment.
6. Discussion mentioned the expected outcomes in terms of integrated management of NCDs within PHC in Kenya and beyond (pages 17-18), I would suggest to add a paragraph or a few lines arguing how this would work in other LMICs with similar contexts or in the same situation.	We agree and we have already included this in the discussion page 17 where we note that the study "will provide important new insights into the requirements for scaling up a novel approach to managing NCDs in primary health care from the perspective of those organising, delivering, and receiving the enhanced services while also expanding our understanding of the unintended consequences of integrating NCD management into the primary care platform on existing care programmes. This knowledge can then be leveraged to inform and improve the design and

	implementation of similar programmes elsewhere in Kenya and other low-resource countries settings.”
--	---

VERSION 2 – REVIEW

REVIEWER	Mahomed, Ozayr University of KwaZulu-Natal, Public Health MEicine
REVIEW RETURNED	10-Jan-2022

GENERAL COMMENTS	Thank you for the revised manuscript. I thank you for the explanation provided. I have the following two points to note:  1. Figure 1 is not enclosed in the revised word document 2. Having now clearly understood the objective 1- my humble opinion is that the articulation of the objective is clearer in the table than the main manuscript. As a qualitative study the word assess depicts a more qualitative measure. It is my opinion that the word explain or explore the quality of leadership and management; levels of stakeholder involvement; adequacy of support mechanisms and resources; ability to adapt the intervention locally; and quality of communication and of monitoring and feedback clearly articulates the objective Although, the authors mention sustainability and refer to it page 7 line 20, the methodology of the protocol does not reflect how the sustainability will be assessed. There are tools such as theNational Health Service (NHS) Institute for Innovation and Improvement Sustainability Model could be easily added to enhance the methods of the study
---

VERSION 2 – AUTHOR RESPONSE

Author response to reviewer comments

1. Figure 1 is not enclosed in the revised word document

Author response: Thank you for highlighting this. We had embedded Figure 1 in our revised document but were advised by the Journal to remove all figures in the main document and upload them separately. We have subsequently resubmitted the (revised) main document and uploaded the figure as a separate pdf file ('bmjopen-2021-056261 Figure 1'). As this is a Journal requirement we are unable to change this.

2a. Having now clearly understood the objective 1- my humble opinion is that the articulation of the objective is clearer in the table than the main manuscript. As a qualitative study the word assess depicts a more qualitative measure. It is my opinion that the word explain or explore the quality of leadership and management; levels of stakeholder involvement; adequacy of support mechanisms and resources; ability to adapt the intervention locally; and quality of communication and of monitoring and feedback clearly articulates the objective

Author response: You recommend we replace the word 'assess' with 'explain' or 'explore' in relation to Objective 1 of our paper, on the basis that the former depicts a more qualitative measure. Since Objective 1 indeed uses entirely qualitative methods, we believe that the term captures our aims well

and we would like to retain it. We have not changed the text

2b. Although, the authors mention sustainability and refer to it page 7 line 20, the methodology of the protocol does not reflect how the sustainability will be assessed. There are tools such as the National Health Service (NHS) Institute for Innovation and Improvement Sustainability Model could be easily added to enhance the methods of the study

Author response: Thank you very much for your suggestion. We believe your comment refers to the following sentence “It [the conceptual framework] posits that ensuring that service innovation, such as the PIC4C model, is sustained, spread and scaled up, requires supportive and committed leadership and management at different tiers; ..”. We should clarify that our study does not seek to assess sustainability as such; instead, as highlighted in the aims and objectives and Table 1 specifically, our study will explore the degree to which the various factors that have been shown in the literature to be important for sustaining and scaling up service innovations are present in PIC4C. The NHS Institute for Innovation and Improvement Sustainability Model is designed to support organisations in understanding the likely sustainability of improvement initiatives by raising awareness of factors considered important for sustainability and encouraging teams to consider actions to increase the likelihood of sustainability. We accept the potential usefulness of this model although note that it has not been widely used, with some authors raising questions about the applicability of the model in practice (see e.g. Doyle et al. 2013; Lennox et al. 2017). The conceptual framework guiding this study is based on a comprehensive review of the extant literature and draws, among others, on seminal work by Greenhalgh et al (2004), which explored understandings and drivers of sustainability of innovations in service delivery and organisation, along with the work by Lennox and colleagues, who have provided extensive reviews around sustainability of health service innovation (e.g. Lennox et al. 2018). We therefore believe that our study will provide insights into the likely sustainability of PIC4C, or elements of it.

VERSION 3 – REVIEW

REVIEWER	Mahomed, Ozayr University of KwaZulu-Natal, Public Health MEDicine
REVIEW RETURNED	08-Feb-2022
GENERAL COMMENTS	The authors have made. case in their rebuttal letter and the differences are merely semantic and will not have a serial impact on the outcome of the study. I will like to suggest that instead of having assess for two objectives consider revising one of the adjectives but that is entirely up to you